# The engaging nature of interactive gestures

Arianna Curioni[1]*, Gunther Klaus Knoblich[1], Natalie Sebanz[1], Lucia Maria Sacheli[2]

1 Department of Cognitive Science, Central European University, Budapest, Hungary, 2 Department of Psychology and Milan Center for Neuroscience (NeuroMi), University of Milano-Bicocca, Milano, Italy

* curionia@ceu.edu

**Data Availability Statement:** All relevant data are within the paper and its Supporting Information files.

**Funding:** European Research Council under the European Union's Seventh Framework Program (FP7/2007-2013) / ERC grant agreement n°

## Abstract

The social interactions that we experience from early infancy often involve actions that are not strictly instrumental but engage the recipient by eliciting a (complementary) response. Interactive gestures may have privileged access to our perceptual and motor systems either because of their intrinsically engaging nature or as a result of extensive social learning. We compared these two hypotheses in a series of behavioral experiments by presenting individuals with interactive gestures that call for motor responses to complement the interaction ('hand shaking', 'requesting', 'high-five') and with communicative gestures that are equally socially relevant and salient, but do not strictly require a response from the recipient ('Ok', 'Thumbs up', 'Peace'). By means of a spatial compatibility task, we measured the interfering power of these task-irrelevant stimuli on the behavioral responses of individuals asked to respond to a target. Across three experiments, our results showed that the interactive gestures impact on response selection and reduce spatial compatibility effects as compared to the communicative (non-interactive) gestures. Importantly, this effect was independent of the activation of specific social scripts that may interfere with response selection. Overall, our results show that interactive gestures have privileged access to our perceptual and motor systems, possibly because they entail an automatic preparation to respond that involuntary engages the motor system of the observers. We discuss the implications from a developmental and neurophysiological point of view.

## Introduction

Gestures are pervasive in our everyday interactions. They are used to communicate and disambiguate meanings (deictic gestures like pointing, symbolic gestures, emblems), to clarify or emphasize discourse (gestures accompanying speech, iconic gestures), and to signify actions (pantomimes). *Communicative gestures* share a common neural substrate with language [1, 2, 3] and constitute precursors of language acquisition, both ontogenetically [4, 5, 6] and phylogenetically [7, 8, 9, 10].

Importantly, there is a specific subset of gestures that not only have the function of transmitting socially relevant information from a communicator to a receiver, but are also interactive, as they call for a specific response in the observer to complete a joint action. For instance, an open palm, depending on its orientation, may call for a hand-shake, a high-five, or a giving action. Here, we hypothesize that the key feature of *interactive gestures* is that they transfer

609819, SOMICS, and by ERC grant agreement n°
616072, JAXPERTISE.

**Competing interests:** The authors have declared
that no competing interests exist.

social information via purely pre-verbal, non-symbolic features that maximally *engage* the
recipient [11, 12]: this would predict that people perceiving interactive gestures may recruit
specific cognitive processes. Interactive gestures may in fact be processed based on a sensori-
motor coding that maps the representation of the observed action onto an interactive script
that triggers a complementary response. This would imply that when observers perceive a ges-
ture that is a component of hand-shaking, a hand-shaking script would be activated in the sen-
sorimotor system of the observer and facilitate the performance of the complementary hand
shaking action (see [13,14,15,16,17,18,19]). In other words, the perception of interactive ges-
tures might entail *social affordances*.

We use the concept of *affordance* in the broad sense that is common in Cognitive Neurosci-
ence studies of action. Traditionally, the term has been used to describe the action possibilities
provided to an organism by the perceptual properties of the environment (e.g., of objects,
[20]). Neuroimaging and lesion studies have corroborated the hypothesis that the perception
of objects induces a behavioural response facilitation; it involves a left-lateralized fronto-parie-
tal network responsible for the preparation of object-specific motor responses, which activates
regardless of the intention to actually interact with the observed object [21, 22, 23, 24]. In a
non-externalist notion of affordances [25, 26, 27], they can thus be considered not only as
object properties, but also as sensorimotor patterns of brain activations that determine specific
behavioural responses. With *social affordances*, then, we refer to the opportunities for social
interactions potentially evoked by interactive gestures, such as the call for a complementary
response that would complete a joint action.

Previous studies have used spatial compatibility tasks to investigate affordance-driven beha-
vioural effects induced by the observation of interactive and communicative gestures. These
tasks provide a measure of the automatic activation of responses that have a spatial overlap
with the spatial feature of the stimuli [28]. When action representations are spontaneously
activated by a visual stimulus, e.g., as a consequence of object affordance, the responses that
are compatible with the evoked action are facilitated (e.g., a left response to a stimulus that is
presented on the left side of the screen), while the responses that are incompatible with the
evoked action show slower response times.

In these studies, interactive gestures have been found to have a facilitatory effect on the
selection of a complementary action [29, 30, 31] as compared to non-social control stimuli (e.
g, wooden hands, [29]), non-social intransitive gestures (e.g., a fist, [31], [29]), or non-social
directional stimuli (e.g. arrows, [30]). For example, [29] report an unexpected reversed com-
patibility effect for an interactive stimulus (a hand-shake, Experiment 1), which primed a com-
plementary and not an imitative response as compared to a non-social stimulus depicting the
same gesture executed by a wooden hand. This effect disappeared if the stimulus was commu-
nicative but not interactive (an OK gesture, Experiment 2), but the two results were not
directly compared. This incidental finding is in line with our prediction that interactive ges-
tures may recruit specific sensorimotor processing routes. However, this and the other previ-
ous studies lacked control stimuli with comparable social relevance, or they averaged
responses to interactive and communicative gestures, thus ignoring potential differences
between these two types of stimuli. Thus, it is an open question whether processing interactive
gestures really involves a different processing route as compared to other socially relevant
stimuli.

Capitalizing on the previous literature, we aimed at directly comparing the response facili-
tation induced by Interactive gestures ('hand shaking', 'requesting', 'high-five') with the
response facilitation induced by Communicative gestures ('Ok', 'Thumbs up', 'Peace') by using
a spatial compatibility task. Participants were instructed to judge target letters based on their
orientation (upright/inverted), while concomitantly observing irrelevant Interactive and

Communicative gestures matched for salience (S1 File). We presented (inter)action-relevant (but task-irrelevant) gestures in a location that was compatible or incompatible with the responding hand and tested whether Communicative and Interactive gestures differently influence participants' accuracy and response speed. We compared the size of the spatial compatibility effect (CE) between the two types of gestures, thus measuring the impact of Interactive vs. Communicative gestures on response selection.

We had the following predictions. First, Communicative gestures should induce a strong CE due to the perceptual salience that characterizes social stimuli, e.g., responding with the left hand should be faster when the Communicative gesture appears in the left visual hemi-field. Second, if Interactive gestures provide a social affordance, they should induce a smaller CE: indeed, the CE may be reduced by the call for a complementary response, e.g., a requesting gesture that appears on the left hemi-field may call for a giving gesture performed with the right hand. In three experiments, we investigated differences in the CE induced by Interactive as compared to Communicative gestures (Experiment 1) and whether they can be truly attributed to social affordance effects (Experiment 2 and 3).

## Experiment 1

In the first experiment, we used a spatial compatibility task [32] to test whether Interactive gestures lead to a reversed or smaller CE than Communicative gestures [29]. This finding would indicate that interactive gestures trigger the representation of complementary responses, i.e., exert social affordances. We also investigated whether the predicted reduction of the CE in the Interactive gesture condition is restricted to the dominant hand, which is the effector normally used in a joint action (e.g., hand shaking with the right hand) or whether it generalizes to the non-dominant hand. Investigating this generalization is crucial to shed light on the processes underlying potential social affordance effects. Indeed, an effect restricted to the dominant hand would indicate that the processing of interactive gestures activates overlearned sensori-motor routes. On the contrary, the generalization of social affordance effects to the non-dominant hand may indicate that they do not depend on the activation of overlearned social scripts and may rather result from intrinsic perceptual features that call for a complementary response, independently of previous experience with the specific social script.

## Methods of Experiment 1

### Participants

We based our sample size on previous studies investigating the modulation of object affordance effects [33, 34], which found a moderate effect size of $\eta_p^2$ = .21 for the interaction between spatial compatibility and the affordance effect. Using a 2x2 within-subject analysis of variance (ANOVA) the power analysis conducted in G*Power 3.1 [35] revealed that, with $\alpha$ = .05 and statistical power at $1-\beta$ = .90, we needed a sample size of N = 27. Thirty participants were recruited to take part in the experiment, of which one was excluded because he did not understand the task instructions, and a second one was excluded because his accuracy and RT data showed a high amount of outliers (see below, final sample 28 participants, 20 f, average age = 26.3 years, SD age = 4.85 years). All experiments in this study were approved by the United Ethical Review Committee for Research in Psychology (EPKEB). All participants reported to be right-handed and to have normal or corrected-to-normal vision. They signed prior informed consent and received monetary compensation. The study was performed in accordance with the Declaration of Helsinki and later amendments.

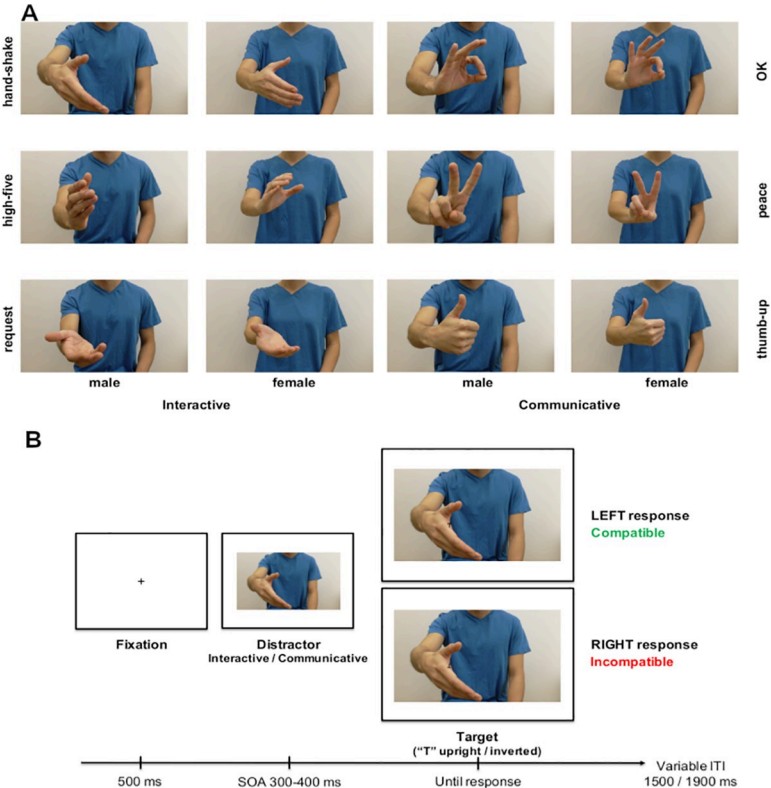

**Fig 1.** A. The images of the gestures used as distractor stimuli. B. The trial-time line. Both experimental stimuli and trial-timeline were identical in Experiment 1, 2 and 3.

## Stimuli and apparatus

The set of stimuli comprised full-colour pictures of interactive and communicative gestures. Each picture was taken to include the torso of the model forward facing (head excluded) and both arms, with only the hand performing the gesture being visible (see Fig 1). The torso occupied the center of the picture so that the hand performing the gesture was lateralised with respect to the model's body and directed towards the observer. For each condition (Interactive and Communicative) there were three different gestures (see Fig 1A), performed by a male and female model. For each gesture, we created a right and left-hand version by mirroring the original picture to avoid any low-level perceptual differences (for a total of 24 stimuli: 3 communicative and 3 interactive gestures performed by a female or male actor and presented in right- and left-hand version). The body stimuli were 500 x 296 pixels, 2.5 cm in height and 4.2 cm in width and subtended 2.6˚ and 4.37˚ of visual angle at a viewing distance of 55 cm.

To verify that stimuli were matched for salience, we ran a preliminary experiment on an independent sample of 15 participants using a go/no-go task. Results indicated no differences in the salience of Interactive vs. Communicative gestures (S1 File).

The experimental script was run and participants' responses were recorded using MatLab 16b software running on a Dell Precision T5610 PC with a screen size of 24 inch and display resolution of 1920 x 1080 at 60Hz.

## Procedure

Each trial started with a fixation cross presented in the center of the screen for 500 ms. The pre-cue stimulus (gesture) appeared in the center of the screen. The Target (a capital letter T, Sans Serif font, size = 18, either upright or inverted) was presented positioned on the model's hand at 90 pixels from the center and at 170 pixels from the top of the image. After a stimulus onset asynchrony (SOA) that randomly varied from 300–400 ms from trial to trial, the Target appeared. The time-interval of the SOA was chosen based on previous findings suggesting that 300 ms is the minimum time required for object affordances to modulate the activation of the motor system [36, 33, 37]. Participants were asked to indicate whether the Target was upright or inverted by pressing one of the two assigned keys with either their left or right index finger (see Fig 1B). The target was displayed until participants had responded, for a maximum of 1.5 s; if a response was not detected within 1.5 s the script proceeded to the next trial. After an inter-trial interval (ITI) that varied randomly between 1500–1900 ms after the response, the next trial started. The keyboard was centered on the computer screen, so that the response keys were lateralized with respect to the stimuli presented. The assigned keys were key A (left side of the keyboard) for left hand responses, and key L (right side of the keyboard) for right hand responses.

## Experimental design

Participants completed 6 experimental blocks of 48 trials each. There were 144 spatially compatible trials and 144 spatially incompatible trials. Spatial compatibility was coded with respect to the combination between the (left/right) hemi-field of the screen where the target letter and the distractor (the hand gesture) appeared and the required (left/right) hand response, which depended on the target orientation (upright/inverted). The distractor and target position were on the same side of the correct hand response (compatible trials) 50% of the time and on the opposite side (incompatible trials) 50%. The association between Target orientation (upright/inverted) and response (left/right) was counterbalanced across participants. Half of the compatible and incompatible trials displayed interactive gestures and half displayed communicative gestures as distractors. The three different gestures in each condition, performed by

**Table 1. Raw group means of Acc and RTs in each experimental condition for the three experiments.**

|  | Response Times (ms) | | Accuracy | |
|---|---|---|---|---|
| **Experiment 1** | *mean* | *standard deviation* | *mean* | *standard deviation* |
| Interactive Compatible | 522 | 68 | 0.98 | 0.03 |
| Interactive Incompatible | 536 | 62 | 0.96 | 0.04 |
| Communicative Compatible | 516 | 66 | 0.97 | 0.02 |
| Communicative Incompatible | 542 | 72 | 0.96 | 0.04 |
| **Experiment 2** |  |  |  |  |
| Interactive Compatible | 531 | 82 | 0.97 | 0.04 |
| Interactive Incompatible | 556 | 85 | 0.98 | 0.05 |
| Communicative Compatible | 527 | 81 | 0.97 | 0.04 |
| Communicative Incompatible | 567 | 87 | 0.98 | 0.04 |
| **Experiment 3** |  |  |  |  |
| Interactive Compatible | 550 | 69 | 0.98 | 0.03 |
| Interactive Incompatible | 570 | 71 | 0.97 | 0.04 |
| Communicative Compatible | 542 | 66 | 0.97 | 0.04 |
| Communicative Incompatible | 587 | 78 | 0.97 | 0.04 |

female and male actors, appeared with equal frequency across trials and were positioned an equal number of times to the right or left side of the screen. Trial order was randomized within and across blocks. Each block took about 4 minutes, leading to 25 minutes to complete the experiment.

## Data analysis

We measured Accuracy (**Acc**), i.e., the proportion of correct responses, and Response Times (**RTs**), i.e., the time delay between the instant when the target letter appeared on the screen and the participants' response, measured on correct trials only. Overall, participants were highly accurate: a response was not detected in 0.73% of trials, equal to 61 trials in the whole sample, and errors were equal to 2.79% of the trials (233 in the whole sample). We planned to exclude participants showing outlier values in both the individual mean **Acc** and individual mean **RTs**, as identified by the Box and Whisker Plot. One participant was excluded from further analysis according to this criterion in Experiment 1.

For illustrative purposes, we report in Table 1A raw **Acc** and **RT** data: here, we calculated the individual mean **Acc** and **RTs** for each condition, excluding from the analysis of **RTs** any outlier values that fell 2.5 SDs above or below the individual mean of each experimental condition.

Data were analyzed in the statistical programming environment R (R 3.3.3, R Core Team 2014). For the analysis of **Acc**, generalized linear mixed effects models were used [38, 39]. As **Acc** is a binary dependent variable, it was submitted to a series of logistic mixed effects regressions using GLMER procedure in "lme4" R package (version 1.1–5, [40]). **RTs** were analyzed as a continuous dependent variable using linear mixed effects models, fitted using the LMER function in "lme4" R package (version 1.1–15, [40]). In both analyses, the inclusion of fixed effects in the best fitting model was tested with a series of likelihood ratio tests, including only the fixed effects that significantly increased the model's goodness of fit [41] (see **S2 Table**). Only the results of the best fitting model are reported.

We considered as fixed effects spatial Compatibility (Compatible vs. Incompatible), Gesture-type (Interactive vs. Communicative), and Response-side (Right hand/Left hand), and their interactions. Concerning the random effect structure, by-subjects and by-stimulus-type (S3 Table) random intercepts were included to account for between-subjects and between-stimuli variability (S2 Table). We report here only the parameters of the best fitting model. In the analysis of **RTs**, we also applied a model criticism procedure to the best fitting model to exclude outlier trials as recommended by [38]. Statistics of the fixed effects of the best fitting model were estimated with the "lmerTest" R package (version 3.0–1, [42]). We report a summary of the fixed effects of the best-fitting models for each variable; for **RTs**, significance levels are based on Satterthwaite's degrees of freedom approximation. When appropriate, the post-hoc direct contrasts between the single levels of the significant interactions and main effects were conducted on the best fitting model with the "phia" R package (version 0.2–1, [43]), applying Bonferroni correction for multiple comparisons. All tests of significance were based upon an α level of 0.05.

## Results of Experiment 1

### Accuracy

The best fitting model only included spatial Compatibility as fixed effect (S2A Table). The results showed a significant main effect of spatial Compatibility (Wald Z = -6.21, $p < 0.001$) indicating that participants were more accurate on Compatible than on Incompatible trials

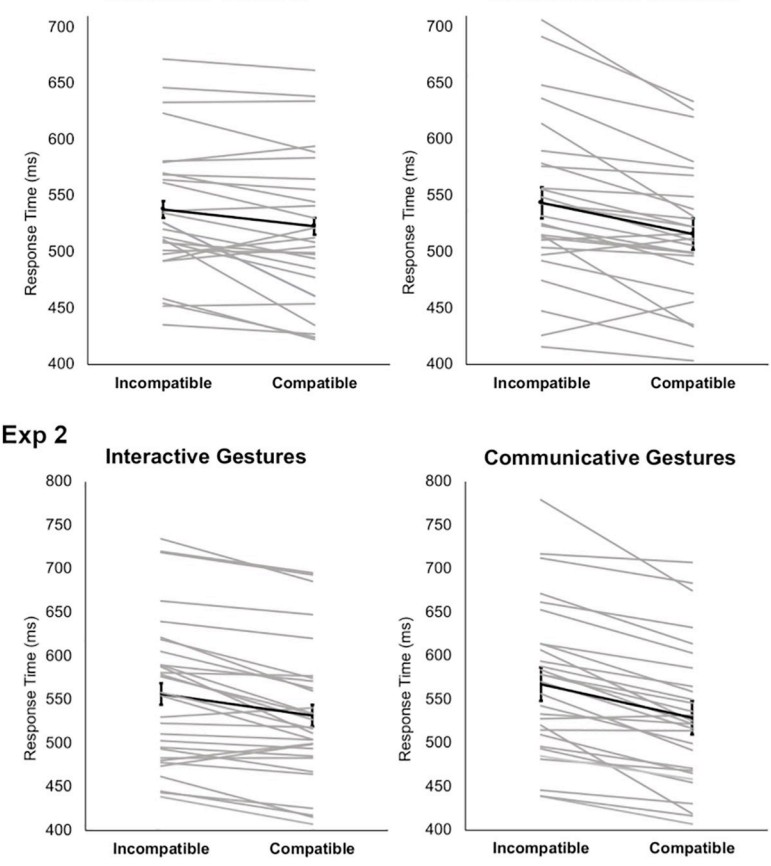

**Fig 2. The figure illustrates the stimulus-type by spatial compatibility interaction effect (CE) that was observed in both Experiment 1 and 2.** Notably, the CE is smaller for Interactive compared to Communicative gestures. Grey lines indicate single-subject values and black thick lines indicate the group means and standard deviations.

(adjusted (adj) mean Compatible trials 0.99, SE 0.23; adj mean Incompatible trials 0.97, SE 0.21).

## Response times

The best fitting model included spatial Compatibility, Gesture-type and Response-side as fixed effects (S2A Table). 3.21% of the trials were excluded from further analysis as outliers (251 trials in the whole dataset). The results showed a significant main effect of spatial Compatibility ($F(1,7516.5) = 117.06$, $p < .001$) and Response-side ($F(1, 7516.3) = 25.46$, $p < .001$), while the main effect of Gesture-type was not significant ($F(1, 10) = 0.15$, $p = .71$). These effects indicate that responses on Compatible trials (adj mean 511.90 ms, SE 10 ms) were faster than on Incompatible ones (adj mean 538.14 ms, SE 10 ms), and responses with the Right hand (adj mean 518.90 ms, SE 10 ms) were faster than with the Left hand (adj mean 531.14 ms, SE 10 ms). Crucially, the results showed a significant Gesture-type x spatial Compatibility interaction ($F(1, 7516.5) = 6.85$, $p = .009$), indicating that the spatial Compatibility effect (CE) for Interactive gestures (Interactive-Compatible, adj mean 516.06 ms, vs. Interactive-Incompatible, adj mean 535.95 ms, $p < .001$) was smaller than for Communicative gestures (Communicative-Compatible, adj mean 507.74 ms, vs. Communicative-Incompatible, adj mean 540.32 ms, $p < .001$).

To directly compare the size of the CE between Interactive and Communicative gestures, we also computed an index of the CE for each participant (RT Incompatible–RT Compatible) separately for Gesture-type. A Dependent Sample t-test revealed a significant difference in the CE between Communicative and Interactive gestures ($t(27)$ = -8.86, $p < 0.001$, $d$ = -1.675), with the effect being bigger for Communicative gestures (mean 17 ms, sd 6 ms) than for Interactive gestures (mean 9 ms, sd 2 ms) (Fig 2).

## Discussion of Experiment 1

In line with our predictions, there was a significant interaction between Gesture-type and spatial Compatibility, indicating that the Interactive gestures led to a reduced CE as compared to Communicative gestures. The reduction of the CE for Interactive vs. Communicative gestures occurred for both the dominant and non-dominant hand. Overall, these results suggest that, in the Interactive gesture condition, the CE induced by the perceptual salience of the social stimuli might be counter-acted by the call for a complementary response, The findings indicate that this effect is not a mere effect of extensive social motor learning as it occurs both in the dominant and non-dominant hand.

## Experiment 2

To further assess whether the reduction of CE in the Interactive gesture condition might be ascribed to a social affordance effect, we asked participants to respond to the target letter with the index and middle finger of their right hand rather than with their left and right hand [44]. Indeed, proper affordance effects trigger a response in an effector-specific way and are thus expected to disappear in intra-manual spatial compatibility tasks [44]. On the contrary, if interactive gestures induce a reduction in the CE in an effector-unspecific way, we expect the reduction in CE to be present even when responses are mapped onto fingers of the same hand: this result would suggest that interactive gestures trigger an automatic preparation to respond that precedes the selection of a specific action program. Finally, in Experiment 2 one might expect a general facilitation for interactive gestures presented in the left hemi-field, i.e., the position that should trigger a response with the right hand used to respond in both Compatible and Incompatible trials: as a consequence, in Experiment 2 the visual hemi-field (left or right) where stimuli appeared was tested as possible predictor of performance in the analyses.

## Methods of Experiment 2

### Participants

Thirty individuals (18 female, average age = 27.03 years, SD age = 4.79 years) participated in Experiment 2. All participants reported to be right-handed and to have normal or corrected-to-normal vision. They signed prior informed consent and received monetary compensation. The study was performed in accordance with the Declaration of Helsinki and later amendments.

### Procedure

The procedure was identical to Experiment 1, except that participants were asked to respond to letter orientation with the index and the middle finger of the right hand. The assigned keys were key J for left side responses with the index finger, and the key L for right side responses with the middle finger.

### Experimental design

The experimental design was the same as in Experiment 1.

### Data analysis

Data analyses were the same as in Experiment 1, with the only difference that we tested the effect of Stimulus-hemifield instead of Response-side. In Experiment 2, the factor Response-side was not tested as possible predictor as we did not expect any difference between the fingers of the same hand. We instead included the factor Stimulus-hemifield in the analysis to investigate whether the side where stimuli appeared could impact on participants' accuracy and response times.

Overall, participants were highly accurate: a response was not detected in 0.66% of trials, equal to 57 trials in the whole sample, and errors were equal to 2.35% of the trials (203 trials in the whole sample). We planned to exclude participants showing outlier values in both the individual mean **Acc** and individual mean **RTs**, as identified by the Box and Whisker plot. No participant was excluded according to this criterion in Experiment 2. Raw **Acc** and **RTs** data are reported in Table 1 and were calculated as described for Experiment 1.

## Results of Experiment 2

### Accuracy

The best fitting model only included spatial Compatibility as fixed effect (S2B Table). The results showed a significant main effect of spatial Compatibility (Wald Z = -8.25, $p < 0.001$), indicating that participants were more accurate on Compatible than on Incompatible trials (adj mean Compatible trials 0.99, SE 0.24; adj mean Incompatible trials 0.97, SE 0.20).

### Response times

The best fitting model included spatial Compatibility and Gesture-type, but not Stimulus-hemifield, as fixed effects (S2B Table). 2.90% of the trials were excluded from further analysis as outliers (242 trials in the whole dataset). The results showed a significant main effect of spatial Compatibility (F(1,8095.5) = 209.48, $p < .001$), while the main effect of Gesture-type was not significant (F(1, 10) = 0.42, $p = .53$). These effects indicate that responses on Compatible trials (adj mean 526.92 ms, SE 13 ms) were faster than on Incompatible ones (adj mean 558.96 ms, SE 13 ms). Crucially, the results showed a significant Gesture-type x spatial Compatibility interaction (F(1, 8095.4) = 12.27, $p < .001$), indicating that the spatial Compatibility effect (CE) for Interactive gestures (Interactive-Compatible, adj mean 529.29 ms, vs. Interactive-Incompatible, adj mean 553.58 ms, $p < .001$) was smaller than for Communicative gestures (Communicative-Compatible, adj mean 524.57 ms, vs. Communicative-Incompatible, adj mean 564.37 ms, $p < .001$).

To directly compare the size of the CE between Gesture-types, we computed an index of the CE for each participant (RT Incompatible–RT Compatible) for Interactive and Communicative gestures. A Dependent Sample t-test revealed a significant difference in the CE between Communicative and Interactive gestures (t(29) = -2.99, $p = 0.006$, $d = -0.547$), with the effect being bigger for Communicative gestures (mean 39 ms, sd 26 ms) than for Interactive gestures (mean 25 ms, sd 24 ms) (Fig 2).

## Discussion of Experiment 2

In Experiment 2 we replicated evidence of a specific reduction in the CE for Interactive as compared to Communicative gestures. This reduction occurred even when the stimulus-

response mapping was intra-manual, and for stimuli presented in both visual hemi-fields: this supports the hypothesis that the reduction of CE for Interactive gestures occurs at a processing level at which the effector performing the possible complementary response is not yet specified. We thus suggest that Interactive gestures might trigger an automatic preparation to act that does not directly translate into a social affordance response, as it does not (only) depend on the activation of specific motor scripts. Importantly, as we found no effect of Stimulus-hemifield, we suggest that performing the task with the effector that is usually involved in the execution of interactive actions (i.e., the right dominant hand) cannot solely account for the pattern of results. To further test this latter point, we aimed to replicate our results in a third experiment where the intra-manual task was performed entirely with the left hand.

## Experiment 3

We designed Experiment 3 to show that the reduction of CE we observed in Experiment 2 was not specific to the right dominant hand, i.e. the effector usually involved in the preparation of the complementary response to an interactive gesture. We thus asked participants to perform the same task as described in Experiment 2 by responding with the index and middle finger of their left hand. If interactive gestures induce a reduction of CE as a result of triggering automatic preparation to respond that occurs at higher levels of motor planning (where the specific movement is not detailed yet), we expect the reduction in CE to be present even when responses are mapped onto fingers of the left hand.

## Methods of Experiment 3

### Participants

We conducted a power analysis in G∗Power 3.1 [35] to determine the sample size required to detect the observed effect size of the Compatibility x Stimulus type second order interaction (i.e., the interaction between compatibility and type of gesture) of $\eta_p^2 = .25$ using a 2x2 within-subject analysis of variance (ANOVA). The analysis revealed that, with $\alpha = .05$ and statistical power at $1–\beta = .80$, we needed a sample size of N = 19. Twenty individuals (14 female, average age = 22.7 years, SD age = 4.36 years) thus participated in Experiment 3. We collected data from one additional participant (total N = 20) to equally balance the stimulus–response mappings between participants. All participants reported to be right-handed and to have normal or corrected-to-normal vision. They signed prior informed consent and received monetary compensation. The study was performed in accordance with the Declaration of Helsinki and later amendments.

### Procedure

The procedure was identical to Experiment 2, except for the fact that participants were asked to perform the task by responding with two fingers of the left hand. Participants were asked to indicate whether the Target was upright or inverted by pressing one of the two assigned keys with their left hand. The assigned keys were key A (middle finger of the left hand) for left side responses, and key D (index finger of the left hand) for right side responses.

### Experimental design

The experimental design was the same as in Experiment 2.

### Data analyses

Data analyses were the same as in Experiment 2. Overall, participants were highly accurate: a response was not detected in 0.85% of trials, equal to 49 trials in the whole sample, and errors occurred on 2.71% of the trials, equal to 156 trials in the whole sample. We planned to exclude participants showing outlier values in both the individual mean **Acc** and individual mean **RTs**, as identified by the Box and Whisker plot. No participant was excluded according to this criterion in Experiment 3.

Raw **Acc** and **RTs** data are reported in Table 1, calculated as described for Experiment 1.

## Results of Experiment 3

### Accuracy

The best fitting model only included spatial Compatibility as fixed effect (S2C Table). The results showed a significant main effect of spatial Compatibility (Wald Z = -5.54, $p < 0.001$), indicating that participants were more accurate on Compatible than on Incompatible trials (adj mean Compatible trials 0.99, SE 0.28; adj mean Incompatible trials 0.97, SE 0.26).

### Response times

The best fitting model included spatial Compatibility and Gesture-type, but not Stimulus-hemifield, as fixed effects (S2C Table). 3.00% of the trials were excluded from further analysis as outliers (167 trials in the whole dataset). The results showed a significant main effect of spatial Compatibility ($F(1,5355.2) = 114.39$, $p < .001$), while the main effect of Gesture-type was not significant ($F(1, 10) = 0.66$, $p = .43$). These effects indicate that responses on Compatible trials (adj mean 544.07 ms, SE 14 ms) were faster than on Incompatible ones (adj mean 576.89 ms, SE 14 ms). Crucially, the results showed a significant Gesture-type x spatial Compatibility interaction ($F(1, 5355.2) = 21.31$, $p < .001$), indicating that the CE for Interactive gestures (Interactive-Compatible, adj mean 547.74 ms, vs. Interactive-Incompatible, adj mean 566.37 ms) was smaller than for Communicative gestures (Communicative-Compatible, adj mean 540.41 ms, vs. Communicative-Incompatible, adj mean 587.40 ms).

To directly compare the size of the CE between Gesture-types, we computed an index of the CE for each participant (RT Incompatible–RT Compatible) for Interactive and Communicative gestures. A Dependent Sample t-test revealed a significant difference in the CE between Communicative and Interactive gestures ($t(19) = -6.0$, $p < 0.001$, $d = -1.342$), with the effect being bigger for Communicative (mean = 42 ms, sd = 13 ms) than for Interactive gestures (mean 18 ms, sd 6 ms).

## Discussion of Experiment 3

Experiment 3 provided evidence that interactive gestures induce a reduction of compatibility effect also in participants performing the task with the left hand. This confirms that perceiving interactive gestures interferes with automatic response selection at a more abstract level of spatial mapping that does not concern the effector used to perform the complementary actions, possibly as a consequence of an automatic (and not yet specified) preparation to respond.

## General discussion

Understanding gestures performed by our conspecifics is fundamental for the development of our social life, as it enables us to timely and successfully engage in interactions, develop and master language, attribute intentions, and take part in collaborative and cultural activities [10]. Previous research on gestures has mainly focused on the shared cognitive processes underlying

the understanding of symbolic gestures and their overlap with other forms of symbolic communication, such as language [1]. Among socially relevant gestures, those that are also interactive, calling for a specific response in the observer to complete a joint action, might constitute privileged stimuli for our perceptual and motor system. Indeed, they potentially *engage* the observer in *actively* taking part in a social interaction. In the present study, we hypothesized that interactive gestures might be processed differently than purely communicative gestures for two possible reasons, either because of their intrinsically engaging nature, or as a result of extensive social learning.

We designed a series of behavioural experiments to test and disentangle these two possibilities by investigating the processing of Interactive vs. Communicative gestures in a spatial compatibility task, where we measured how response selection was modulated by the (task-irrelevant) gestures presented in a compatible or incompatible position with respect to participants' response hand.

The results of the first experiment show that the perception of Interactive gestures leads to a reduced spatial compatibility effect (CE) compared to Communicative gestures. This result supports the hypothesis that different cognitive processes mediate the perception of Interactive and Communicative gestures. As stimuli in the two categories were matched for perceptual salience (S1 File) and social relevance, we can rule out low-level explanations for our pattern of results. The effect we found is in line with evidence of "interference" effects generated by perceived interactive gestures on the execution of pre-planned non-interactive movements [45], and it may seem to be driven by the activation of an interactive and complementary action script (i.e. a social affordance effect). However, the results of the first experiment provide indication that this modulation can be observed when responding with both the dominant and non-dominant hand. This raises the possibility that the effect we observe does not originate from an affordance effect due to extensive social motor learning of a complementary response (e.g. learning to shake hands with the right-dominant hand), but instead relies on an automatic motor preparation preceding the selection of a specific action program. This would suggest that, rather than affording a response to complement a specific joint action script, interactive gestures might produce a more generic motor engagement that prepare us to (inter)act.

The results of our second and third experiments support this latter interpretation. Indeed, they replicate the pattern of results of the first experiment and indicate that the selective reduction of CE for Interactive gestures occurred even when the stimulus-response mapping was intra-manual, when stimuli were presented in both visual hemi-fields, and regardless of whether participants performed the task with their right or left hand. Altogether, the pattern of results of these two additional experiments may seem to be incompatible with the notion of affordance-based motor preparation; however, it has to be noted that our stimuli set was composed of three different Interactive gestures (all requiring dominant hand responses), therefore it is possible that we observed a modulation resulting from the preparation of multiple motor plans, all relevant and coherent to the target goal of completing a joint action. Recent affordance models argue that the process of selecting a motor plan may occur simultaneously for multiple actions relevant to the target via attentional mechanisms (i.e. affordance competition hypothesis, [46]). If multiple motor plans are activated concurrently by the same perceived affordance, it might not be possible to observe a behavioral facilitation or modulation that is specific to any of the motor plans in the very early stages of processing. Furthermore, there is evidence that affordance-like effects are modulated by (other than motor) cognitive processes such as attention allocation [37], and that effector selection and action specification may emerge at late stages of action planning [36, 47, 48].

We suggest that the most parsimonious interpretation of our results is that interactive gestures might directly *engage* the observer and produce a readiness to interact, which does not

(yet) result in the preparation of the specific complementary response. At a neural level, this might entail the automatic recruitment of premotor neural resources involved in response preparation that precede the selection of the effector used to act. By analogy with object affordance, these resources might include the anterior pre-supplementary motor cortex (pre-SMA) that typically codes a yet unspecified readiness for action associated with the perception of affordable targets [49]. These properties of pre-SMA have been widely described by studies in non-human primates [50, 51] and seem to also play an important role in social learning [52, 53]. More broadly, the results of our experiments clearly indicate that Interactive gestures are processed via a different route as compared to equally salient and socially relevant gestures that do not have an affordance for motor interaction. We speculate that, while being the recipient of Communicative gestures that carry symbolic meaning activates semantic processing routes [54], the encoding of Interactive gestures might instead rely on preverbal, non-symbolic/semantic mechanisms associated with earlier developing and more direct processing routes. Therefore, in addition to the recruitment of pre-SMA, the engaging nature of interactive gesture may also depend on the involvement of the fronto-striatal system, which plays a crucial role when people perceive that others are responsive to their social cues [11, 55].

The capacity of detecting possibilities for joint actions with conspecifics is fundamental to develop social and coordination skills [56] and fast discern actions that might require a social response. As such, this ability might be early acquired while infants take part in interactions mediated by some form of infant-directed interactive gesture, allowing them to parse interactive gestures as part of social interaction scripts (i.e. the action script of giving and requesting, see for instance [57]). The results of the present study indicate that the involuntary perception of interactive gestures influences the participants' responses, as if our perceptual system was equipped with the ability to fast identify opportunity for interactions independently of the actual engagement in a social exchange: these findings pave the way for future investigations to address when the "perceptual advantage" of interactive gestures emerges and what social abilities it might require.

## Supporting information

**S1 Table. Average response times and accuracy per stimulus at the Go/No-Go task.**
(DOCX)

**S2 Table. Model selection for the analysis of accuracy and reaction times in each experiment.**
(DOCX)

**S3 Table. The stimulus-type variable that was used as random intercept to account for the between-stimuli variability.**
(DOCX)

**S1 Data.**
(XLSX)

**S2 Data.**
(XLSX)

**S3 Data.**
(XLSX)

**S1 File.**
(DOCX)

## Author Contributions

**Conceptualization:** Arianna Curioni, Gunther Klaus Knoblich, Natalie Sebanz, Lucia Maria Sacheli.

**Data curation:** Arianna Curioni.

**Formal analysis:** Lucia Maria Sacheli.

**Investigation:** Arianna Curioni.

**Supervision:** Gunther Klaus Knoblich, Lucia Maria Sacheli.

**Writing – original draft:** Arianna Curioni.

**Writing – review & editing:** Arianna Curioni, Gunther Klaus Knoblich, Natalie Sebanz, Lucia Maria Sacheli.

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
