## [Decision Letter · Decision Letter 0]

18 Mar 2020

PONE-D-20-02355

The engaging nature of interactive gestures

PLOS ONE

Dear Dr. curioni,

Thank you for submitting your manuscript to PLOS ONE. After careful consideration, we feel that it has merit but does not fully meet PLOS ONE’s publication criteria as it currently stands. Therefore, we invite you to submit a revised version of the manuscript that addresses the points raised during the review process.

As noted by both reviewers, this is a very well-written manuscript and the findings are an important contribution to the current literature on interactive gestures. Please make the suggested changes, including the additional statistical analyses as suggested by Reviewer #2. Also, please incorporate the citations that both reviewers suggest.

We would appreciate receiving your revised manuscript by May 02 2020 11:59PM. To enhance the reproducibility of your results, we recommend that if applicable you deposit your laboratory protocols in protocols.io, where a protocol can be assigned its own identifier (DOI) such that it can be cited independently in the future. For instructions see: http://journals.plos.org/plosone/s/submission-guidelines#loc-laboratory-protocols

We look forward to receiving your revised manuscript.

Kind regards,

Julie Jeannette Gros-Louis, PhD

Academic Editor

PLOS ONE

Journal Requirements:

Reviewers' comments:

Reviewer's Responses to Questions

**Comments to the Author**

1. Is the manuscript technically sound, and do the data support the conclusions?

Reviewer #1: Yes

Reviewer #2: Yes

2. Has the statistical analysis been performed appropriately and rigorously? 

Reviewer #1: Yes

Reviewer #2: Yes

3. Have the authors made all data underlying the findings in their manuscript fully available?

Reviewer #1: Yes

Reviewer #2: Yes

4. Is the manuscript presented in an intelligible fashion and written in standard English?

Reviewer #1: Yes

Reviewer #2: Yes

5. Review Comments to the Author

Reviewer #1: This extremely well-written ms. describes an elegant suite of studies that explore the engaging nature of "interactive" rather than "communicative" actions, i.e. actions which require activity on the part of the human observer drawing him/her in to interact. By cleverly combining this with a spatial compatibility task, the interference of such stimuli was objectively assessed. Across different experiments it was shown that interactive gestures impact response selection and spatial compatibility as compared to communicative actions. The authors interprete this as evidence for a priviliged access of interactive gestures to perceptual and action systems.

I applaud the authors to systematically addressing this subtle, but important distinction! I am very impressed by the choice of stimuli and the methodological rigor with which the studies were conducted. I strongly recommend publication of this ms. and only have minor comments:

1) A study by von der Lühe et al. (2016) also investigated the impact of "communicative" actions. The authors may want to make reference to this study and could also use this opportunity to briefly discuss whether interactive gestures might bring about different expectations/predictions in the observer and how this might play out in a predictive coding scheme.

2) Schilbach and colleagues published several papers (2012, 2013), in which they investigated the impact of social gaze cues on spatial compatibility effects. This papers appear to be missing in the bibliography.

3) Finally, it might be tempting to speculate how interactive gestures might be important (motivationally and otherwise) for matters of mental health. It has been suggested that psychiatric disorders could be construed as "disorders of social interaction", where interactive gestures might also figure prominently (Schilbach 2016). Maybe the authors might want to speculate (or not).

Reviewer #2: Curioni and co-authors report an interesting study in which, in three behavioral experiments, they tested the difference between two types of social actions, namely interactive and communicative gestures, in a spatial compatibility task. By measuring the interference effect of observing lateralized task-irrelevant gestures when participants were requested to respond to target stimuli, they showed a reduced spatial compatibility effect in reaction times for interactive gestures (which were chosen to elicit a complementary response in the observers) as compared to communicative gestures, irrespective of the responding hand. Response accuracy showed that participants were more accurate when the position of the distractor and target was on the same side, regardless of the type of stimulus (interactive, communicative).

The manuscript is well written and provides an interesting extension to the available literature. The experimental design is sound and the data were rigorously analyzed.

I have few queries that the Authors should address to improve the clarity of the manuscript.

1) The first point regards the discussion of the obtained results with reference to the potential difference in salience for interactive gestures, given their intrinsically engaging nature, as compared to communicative gestures.

The results of all three experiments consistently show for RTs that interactive gestures lead to a reduction of the spatial compatibility effect as compared to communicative gestures. Therefore, it emerged that for interactive gestures, incompatible trials (distractor and targets on opposite sides) were less incompatible/interfering than for communicative gestures. Given that this effect was not limited to the dominant hand which would have been recruited to perform a complementary response to the presented interactive gesture, the Authors conclude that this could seemingly be a generalized effect. Thus, the effect would not reflect any specification of the required motor plan, but be related to a general motor engagement to socially interact which is activated when we are faced with interactive gestures. This unspecific effect for interactive gestures could be dependent on differences in the adopted types of stimuli, however the Authors report to have controlled for this in a go/no-go control task reported in the supplementary materials (e.g., p.18, line 423: “stimuli in the two categories were matched for perceptual salience and social relevance”). Nonetheless, it is unclear to me how the adopted go/no-go task can disentangle the issue of difference in salience/social relevance between communicative and interactive gestures.

I would then ask the Authors to comment on this aspect as it is relevant for the interpretation of the obtained results. In addition, please better report both in the supplementary materials and in the main text the rationale of having used this type of control task for the validation of the experimental stimuli and to control for perceptual salience and social relevance.

2) In addition, the statistical analysis of the go/no-go task included only the Stimulus number as within-subject factor, however it would be relevant to test the role of Gesture-type to be able to differentiate between the two types of gestures. Plus, in the control task the results for the accuracy are missing, while only the ANOVA results for RTs are reported. Please, add these missing information to the results paragraph (see supplementary materials).

3) Results: please report a measure of effect size in the results.

In relation to this, is the ηp 2 = .25 reported in the sample size calculation (pp.14-15, lines 350-355) as the observed effect size of the Compatibility x Stimulus type second order interaction referring to results of Experiment 1 or 2?

4) Discussion: When discussing the obtained results in the Discussion paragraphs, please specify that they refer to RTs and do not apply also to accuracy data (e.g., p. 11, lines 257-263; “there was a significant interaction between Gesture-type and spatial Compatibility, indicating that the Interactive gestures led to a reduced CE as compared to Communicative gestures”).

5) When discussing the role of interference effects in interactive situations (e.g., p.18, lines 424-427), the Authors may also find of interest a recent paper in which the observation of an interactive request gesture is reported to facilitate the execution of an incongruent, but appropriate, complementary response, while interfering with a congruent, but inappropriate, action (Betti et al., 2019, PeerJ).

Typo: a dot is missing in p.11, line 262.

6. PLOS authors have the option to publish the peer review history of their article (what does this mean?). If published, this will include your full peer review and any attached files.

Reviewer #1: No

Reviewer #2: No

---

## [Author Response · Author response to Decision Letter 0]

31 Mar 2020

Dear Dr. Gros-Louis,

We would like to thank you for giving us the opportunity to submit a revised version of our manuscript entitled “The engaging nature of interactive gestures” and the Reviewers for their thoughtful comments. We have addressed all points raised by the Reviewers. In the point-by-point response letter, the Reviewer’s points are reported in plain font, our replies in bold font and preceded by "A.R." (Authors’ Response). Changes to the main text are highlighted in italics.

We hope that you and the Reviewers will be satisfied with our revisions.

Kind regards

Arianna Curioni, also on behalf of Günther Knoblich, Natalie Sebanz and Lucia Maria Sacheli

 

Reviewers’ comments

Reviewer #1: This extremely well-written ms. describes an elegant suite of studies that explore the engaging nature of "interactive" rather than "communicative" actions, i.e. actions which require activity on the part of the human observer drawing him/her in to interact. By cleverly combining this with a spatial compatibility task, the interference of such stimuli was objectively assessed. Across different experiments it was shown that interactive gestures impact response selection and spatial compatibility as compared to communicative actions. The authors interprete this as evidence for a priviliged access of interactive gestures to perceptual and action systems.

I applaud the authors to systematically addressing this subtle, but important distinction! I am very impressed by the choice of stimuli and the methodological rigor with which the studies were conducted. I strongly recommend publication of this ms. and only have minor comments:

A.R. We thank Reviewer #1 for her positive evaluation of our manuscript and for the constructive comments that helped us to improve our manuscript. Please find our point by point response below.

1) A study by von der Lühe et al. (2016) also investigated the impact of "communicative" actions. The authors may want to make reference to this study and could also use this opportunity to briefly discuss whether interactive gestures might bring about different expectations/predictions in the observer and how this might play out in a predictive coding scheme.

A.R. We thank Reviewer #1 for offering this interesting discussion point. We now refer to this study in the Introduction, and have amended the main text as follows (page 3):

“It has been already proposed, in fact, that we might process social actions in terms of their sensorimotor consequences in the world, as this would give us a predictive advantage and the possibility to prepare for a timely response (Kilner et al,. 2007; von der Lühe 2016).”

2) Schilbach and colleagues published several papers (2012, 2013), in which they investigated the impact of social gaze cues on spatial compatibility effects. This papers appear to be missing in the bibliography.

A.R. We thank Reviewer #1 for pointing out this gap in our reference list. We agree that the Schilbach et al. studies are indeed relevant for the present manuscript. We have therefore added a reference to these important papers in the introduction (page 4):

“Similar facilitatory effects have been also found in spatial compatibility tasks where social gaze cues were presented to participants prior to the go signal and reduced the compatibility effect (Schilbach et al., 2012, 2011)”

And discuss the present findings in light of the Schilbach et al. studies (page 19):

“Interesting, this would be in line with evidence of a similar facilitatory effect recorded during spatial compatibility tasks when social gaze cues were presented to participants: in a series of studies, Schilbach at al. showed that being ‘looked at’ by a virtual other induces a social engagement that facilitates the selection of the spatially incongruent response, resulting in a reduction of the compatibility effect (Schilbach et al., 2012, 2011).

3) Finally, it might be tempting to speculate how interactive gestures might be important (motivationally and otherwise) for matters of mental health. It has been suggested that psychiatric disorders could be construed as "disorders of social interaction", where interactive gestures might also figure prominently (Schilbach 2016). Maybe the authors might want to speculate (or not).

A.R. We thank Reviewer #1 for raising the interesting point of the possibility that several psychiatric conditions could be conceived at disorders of social cognition impairments (defined as the impairment of the processes involved in understanding and engaging with, rather than merely observing, other people). Notwithstanding the relevance of this topic for understanding the behavioral and neural processes supporting social interactions, we feel that our work does not directly contribute to this debate as we do not compare data from neurotypical and clinical populations. However, we have mentioned at the end of the Discussion (page 20) that the application of our paradigm to psychiatric conditions may be an interesting topic for future studies:

“The results of the present study indicate that the involuntary perception of interactive gestures influences the participants’ responses, as if our perceptual system was equipped with the ability to fast identify opportunity for interactions independently of the actual engagement in a social exchange: these findings pave the way for future investigations to address when the "perceptual advantage" of interactive gestures emerges, what social abilities it might require, and whether it might be impaired in situations where the possibility to engage in successful interactions is also impaired, as it might be the case in psychiatric disorders [62]”

 

Reviewer #2: Curioni and co-authors report an interesting study in which, in three behavioral experiments, they tested the difference between two types of social actions, namely interactive and communicative gestures, in a spatial compatibility task. By measuring the interference effect of observing lateralized task-irrelevant gestures when participants were requested to respond to target stimuli, they showed a reduced spatial compatibility effect in reaction times for interactive gestures (which were chosen to elicit a complementary response in the observers) as compared to communicative gestures, irrespective of the responding hand. Response accuracy showed that participants were more accurate when the position of the distractor and target was on the same side, regardless of the type of stimulus (interactive, communicative).

The manuscript is well written and provides an interesting extension to the available literature. The experimental design is sound and the data were rigorously analyzed.

I have few queries that the Authors should address to improve the clarity of the manuscript.

A.R. We thank Reviewer #2 for her positive evaluation of our manuscript and for the constructive comments that helped us improve our manuscript. Please find our point by point response below.

1) The first point regards the discussion of the obtained results with reference to the potential difference in salience for interactive gestures, given their intrinsically engaging nature, as compared to communicative gestures.

The results of all three experiments consistently show for RTs that interactive gestures lead to a reduction of the spatial compatibility effect as compared to communicative gestures. Therefore, it emerged that for interactive gestures, incompatible trials (distractor and targets on opposite sides) were less incompatible/interfering than for communicative gestures. Given that this effect was not limited to the dominant hand which would have been recruited to perform a complementary response to the presented interactive gesture, the Authors conclude that this could seemingly be a generalized effect. Thus, the effect would not reflect any specification of the required motor plan, but be related to a general motor engagement to socially interact which is activated when we are faced with interactive gestures. This unspecific effect for interactive gestures could be dependent on differences in the adopted types of stimuli, however the Authors report to have controlled for this in a go/no-go control task reported in the supplementary materials (e.g., p.18, line 423: “stimuli in the two categories were matched for perceptual salience and social relevance”). Nonetheless, it is unclear to me how the adopted go/no-go task can disentangle the issue of difference in salience/social relevance between communicative and interactive gestures.

I would then ask the Authors to comment on this aspect as it is relevant for the interpretation of the obtained results. In addition, please better report both in the supplementary materials and in the main text the rationale of having used this type of control task for the validation of the experimental stimuli and to control for perceptual salience and social relevance.

A.R. We agree with Reviewer #2 that it is important to address the rationale of the control experiment more extensively. For the Communicative gestures set, we have chose stimuli that carry a positive/engaging social meaning for the observer, and that are performed towards the observer. In order to ensure that the two stimuli sets (Interactive and Communicative) did not differ in terms of perceptual salience, we ran a control task where we measured the time and accuracy in detecting each gesture when randomly presented together with all the other gestures. The results showed no difference in response time and accuracy among individual stimuli, and also show no overall difference between stimulus sets (Interactive and Communicative, see response to the Reviewer’s Point N.2). We chose such a perceptual task because it provides an independent behavioural measure of the difficulty of stimulus detection. We preferred this over an explicit validation of the stimuli set, that may have potentially been biased by interindividual differences in interpreting the questions and/or the rating scale. We also would like to point out that, in order to further control for potential differences driven by single stimuli within the two sets, we performed all main analyses including stimulus number in the random structure, so that inter-stimulus differences (e.g., in salience) were controlled for. Importantly, moreover, we would like to drive the reviewer’s attention on the fact that we never find a significant main effect of gesture-type: this already indicates that any alternative explanation of our results in terms of perceptual or salience difference between the two categories (Interactive vs. Communicative) cannot account for our results. We have now clarified the rationale at the basis of our go-no go task in the Method section (page 7):

“To verify that stimuli were matched for salience, we ran a preliminary experiment on an independent sample of 14 participants using a go/no-go task. In this task, we measured the time and accuracy in detecting each gesture of the stimulus set when randomly presented together with all the gestures of the set. We chose such a perceptual task as it provides an independent behavioural measure of stimulus detection. If gestures are matched for salience and valence, detection performance (accuracy and response times) should be comparable across gesture categories and across individual gestures. We preferred this over an explicit validation of the stimuli set because such validations may be influenced by interindividual differences in interpreting the validation questions, and/or rating scales. The results indicated that the salience of Interactive and Communicative gestures was comparable, as this was also supported by a Bayesian statistical analysis (see Supplementary Materials).”

We have also amended the Supplementary material accordingly.

2) In addition, the statistical analysis of the go/no-go task included only the Stimulus number as within-subject factor, however it would be relevant to test the role of Gesture-type to be able to differentiate between the two types of gestures. Plus, in the control task the results for the accuracy are missing, while only the ANOVA results for RTs are reported. Please, add these missing information to the results paragraph (see supplementary materials).

A.R. We are grateful to the Reviewer for pointing out these omissions. We now report in the Supplementary material a table summarizing the RTs and ACC results of the control task (see Supplementary Table S1). We also compared the mean ACC and RTs at the go/no go task between the two stimulus categories by means of a non-parametric test (for ACC data) and a paired-sample t-test (for RTs data). These analyses revealed no difference between categories, as also confirmed by a Bayesian comparison (see Supplementary Results):

“We also compared the mean Accuracy (Acc) and Reaction Times (RTs) between Stimulus Category (Interactive, mean Acc 0.88 +/- 0.12, mean RTs 645.32 +/- 85.37; Communicative, mean Acc 0.91 +/- 0.10, mean RTs 639.58 +/- 90.30). The non-parametric Wilcoxon test comparing mean Acc between Stimulus Categories showed no significant effect (W = 41.5, p = 0.17). The paired-sample t-test comparing mean RTs also showed no significant effect (t(13) = -.57, p = .58). 

Finally, we performed a Bayesian Paired-Sample T-Test on RTs data to explore whether the data provided evidence in favor of the null hypothesis (i.e., absence of difference between Stimulus categories). The results showed a Bayesian Factor (BF10) equal to 0.31, indicating a moderate evidence in favor of the null hypothesis, i.e., that the mean RTs showed by the participants at the control task were equal for Communicative and Interactive gestures, suggesting there was no difference in salience between the Stimulus categories.”

3) Results: please report a measure of effect size in the results.

In relation to this, is the ηp 2 = .25 reported in the sample size calculation (pp.14-15, lines 350-355) as the observed effect size of the Compatibility x Stimulus type second order interaction referring to results of Experiment 1 or 2?

A.R. We thank the reviewer for pointing out that the description of the power analysis was unclear. The Power analysis run to establish the required sample-size for the Experiment 3 was indeed based on the results from Experiment 2, as both experiments use an intra-manual task. This point is now clarified at page 15. In this regard, we highlight that all the results reported in the main text are based on mixed model analyses, as indicated in the methods section: we believe this analytical approach is crucial in our design to control for between-subject and between-stimulus variability. Yet, the quantitative estimation of effect size directly from mixed-model analyses is quite problematic. Thus, in order to run the power analysis with G-Power, we re-analyzed the data of the Experiment 2 with a GLM: this analysis showed a significant Stimulus category by Compatibility interaction (F(1,29)= 8.96, p = .006) with an ηp 2 = .24 (this value has been amended, we apologize for the typo).

Moreover, to provide a clear indication of the effect-size, in each experiment we calculated the Cohen’s d values of each effect of interest based on the marginal means and standard deviations of the compatibility effect. These results are the following:

Experiment 1, d = -1.675 (see page 11):

“To directly compare the size of the CE between Interactive and Communicative gestures, we also computed an index of the CE for each participant (RT Incompatible – RT Compatible) separately for Gesture-type. A Dependent Sample t-test revealed a significant difference in the CE between Communicative and Interactive gestures (t(27) = -8.86, p < 0.001, d = -1.675), with the effect being bigger for Communicative gestures (mean 17 ms, sd 6 ms) than for Interactive gestures (mean 9 ms, sd 2 ms) (Fig 2).”

Experiment 2, d = -0.547 (page 14):

“To directly compare the size of the CE between Gesture-types, we computed an index of the CE for each participant (RT Incompatible – RT Compatible) for Interactive and Communicative gestures. A Dependent Sample t-test revealed a significant difference in the CE between Communicative and Interactive gestures (t(29) = -2.99, p = 0.006, d = -0.547), with the effect being bigger for Communicative gestures (mean 39 ms, sd 26 ms) than for Interactive gestures (mean 25 ms, sd 24 ms) (Figure 2).”

Experiment 3, d = -1.342 (page 17):

“To directly compare the size of the CE between Gesture-types, we computed an index of the CE for each participant (RT Incompatible – RT Compatible) for Interactive and Communicative gestures. A Dependent Sample t-test revealed a significant difference in the CE between Communicative and Interactive gestures (t(19) = -6.0, p < 0.001, d = -1.342), with the effect being bigger for Communicative (mean = 42 ms, sd = 13 ms) than for Interactive gestures (mean 18 ms, sd 6 ms).”

4) Discussion: When discussing the obtained results in the Discussion paragraphs, please specify that they refer to RTs and do not apply also to accuracy data (e.g., p. 11, lines 257-263; “there was a significant interaction between Gesture-type and spatial Compatibility, indicating that the Interactive gestures led to a reduced CE as compared to Communicative gestures”).

A.R. We have amended the main text accordingly.

5) When discussing the role of interference effects in interactive situations (e.g., p.18, lines 424-427), the Authors may also find of interest a recent paper in which the observation of an interactive request gesture is reported to facilitate the execution of an incongruent, but appropriate, complementary response, while interfering with a congruent, but inappropriate, action (Betti et al., 2019, PeerJ).

A.R. We now discuss this study on page 18:

“Our results are also in line with evidence for a facilitation of socially relevant motor responses during the observation of interactive requests, even when such responses are kinematically incongruent with the observed actions: this suggests that social affordance effects are strong enough to override the impact of automatic imitation of observed actions [Betti et al., 2019]. “

Typo: a dot is missing in p.11, line 262.

A.R. We have amended the main text accordingly.

---

## [Editor Report · Decision Letter 1]

8 Apr 2020

The engaging nature of interactive gestures

PONE-D-20-02355R1

Dear Dr. curioni,

We are pleased to inform you that your manuscript has been judged scientifically suitable for publication and will be formally accepted for publication once it complies with all outstanding technical requirements.

With kind regards,

Julie Jeannette Gros-Louis, PhD

Academic Editor

PLOS ONE
---

## [Editor Report · Acceptance letter]

13 Apr 2020

PONE-D-20-02355R1 

The engaging nature of interactive gestures 

Dear Dr. curioni:

I am pleased to inform you that your manuscript has been deemed suitable for publication in PLOS ONE. Congratulations! Your manuscript is now with our production department. 

With kind regards,

on behalf of

Dr. Julie Jeannette Gros-Louis 

Academic Editor

PLOS ONE